# Mesenchymal Stem Cell Exosomes Enhance Posterolateral Spinal Fusion in a Rat Model

**DOI:** 10.3390/cells13090761

**Published:** 2024-04-29

**Authors:** Wing Moon Raymond Lam, Wen-Hai Zhuo, Long Yang, Rachel Tan, Sai Kiang Lim, Hwee Weng Dennis Hey, Wei Seong Toh

**Affiliations:** 1Department of Orthopaedic Surgery, Yong Loo Lin School of Medicine, National University of Singapore, 1E Kent Ridge Road, Singapore 119228, Singapore; 2Tissue Engineering Program, Life Sciences Institute, National University of Singapore, 27 Medical Drive, Singapore 117510, Singapore; 3Institute of Molecular and Cell Biology, Agency for Science, Technology and Research (A*STAR), 61 Biopolis Drive, Proteos, Singapore 138673, Singapore; lim_sai_kiang@imcb.a-star.edu.sg; 4Integrative Sciences and Engineering Program, NUS Graduate School, National University of Singapore, 21 Lower Kent Ridge Road, Singapore 119077, Singapore

**Keywords:** exosomes, extracellular vesicles, mesenchymal stem cells, spinal fusion, bone

## Abstract

Spinal fusion, a common surgery performed for degenerative lumbar conditions, often uses recombinant human bone morphogenetic protein 2 (rhBMP-2) that is associated with adverse effects. Mesenchymal stromal/stem cells (MSCs) and their extracellular vesicles (EVs), particularly exosomes, have demonstrated efficacy in bone and cartilage repair. However, the efficacy of MSC exosomes in spinal fusion remains to be ascertained. This study investigates the fusion efficacy of MSC exosomes delivered via an absorbable collagen sponge packed in a poly Ɛ-caprolactone tricalcium phosphate (PCL–TCP) scaffold in a rat posterolateral spinal fusion model. Herein, it is shown that a single implantation of exosome-supplemented collagen sponge packed in PCL–TCP scaffold enhanced spinal fusion and improved mechanical stability by inducing bone formation and bridging between the transverse processes, as evidenced by significant improvements in fusion score and rate, bone structural parameters, histology, stiffness, and range of motion. This study demonstrates for the first time that MSC exosomes promote bone formation to enhance spinal fusion and mechanical stability in a rat model, supporting its translational potential for application in spinal fusion.

## 1. Introduction

Spinal fusion is a common surgical treatment for degenerative lumbar conditions, with over 450,000 operations performed annually in the United States alone [1]. While this surgical treatment is generally effective in relieving pain from neural compression or spinal instability, 5 to 15% of patients may not achieve fusion [2,3]; moreover, this rate could increase up to 50% in revision [4] and multi-level fusion surgeries [5], and in the presence of other patient-related risk factors and comorbidities. Patients with pseudoarthrosis can suffer from long-standing back and leg pain leading to disability and poor quality of life [6,7].

The use of bone graft and its substitutes is imperative in spinal fusion. While autologous bone graft is effective and long regarded as the “gold standard” in spinal fusion, its use is limited by tissue availability and donor site morbidity [8,9]. On the other hand, bone graft substitutes such as demineralized bone matrices are not limited by availability but have variable osteogenic properties and give highly unreliable fusion rates [10]. This has led to the use of osteoinductive factors such as recombinant human bone morphogenetic protein 2 (rhBMP-2) that is applied to an absorbable collagen sponge (ACS) (Infuse^®^, Medtronic Sofamor Danek, Memphis, TN, USA) to stimulate bone formation and facilitate fusion. Despite having an excellent fusion rate of more than 90% [11], the use of rhBMP-2 has been associated with various adverse effects such as radiculitis [12], heterotopic ossification [13] and seroma [14]. Other adverse complications such as dysphagia and airway compression were also reportedly associated with the use of rhBMP-2, and this led to a public health notification from the US Food and Drug Administration (FDA) in 2008 against its application in cervical fusion [15]. Recent research efforts have thus steered towards the development and use of biologics such as mesenchymal stromal/stem cells (MSCs) as an adjunct to promote bone growth and enhance spinal fusion. To date, MSCs in combination with various bone graft substitutes have achieved promising outcomes in both animal and clinical studies [16,17,18]. However, implementation of cell-based MSC therapies is hampered by challenges in maintaining the optimal cell viability and vitality from manufacturing and storage to final patient delivery.

It is now well accepted that MSCs mediate tissue repair through the secretion of trophic factors [19]. Among the trophic factors, extracellular vesicles (EVs) of size 50–200 nm, which include exosomes, have been identified as the principal agents underpinning the therapeutic efficacy of MSCs in several injuries and diseases, including musculoskeletal disorders [20,21]. Specifically, we and others have reported that MSC exosomes were efficacious in repairing osteochondral lesions and bone defects in animal models [22,23,24,25]. However, it remains to be determined if such efficacy extends to enhanced spinal fusion. Unlike the repair of bone defects such as those of the calvaria, the process of new bone formation at the posterolateral spinal fusion site is affected by the spinal microenvironment and biomechanics [26]. The spinal fusion site represents one of the most challenging sites of bone healing, due to the inherent needs of the spine to support a wide range of motion, with which high strains or unwanted motion can lead to undesirable fibrous tissue formation and non-union [27].

Therefore, this study aimed to investigate the use of MSC exosomes in spinal fusion. To our knowledge, the effects of MSC exosomes on spinal fusion have not been investigated. We hypothesized that MSC exosomes could promote bone formation to enhance spinal fusion. With the use of an established rat posterolateral spinal fusion model, the effects of a single implantation of exosome-supplemented scaffold on spinal fusion were evaluated.

## 2. Materials and Methods

### 2.1. Preparation and Characterization of MSC Exosomes

MSC exosomes were prepared from an immortalized E1-MYC 16.3 human MSC line [28]. For exosome preparation, the conditioned medium was prepared by growing 80% confluent cells in a chemically defined medium composed of Dulbecco’s Modified Eagle Medium (DMEM) supplemented with 1% nonessential amino acids, 1% glutamine, 1% insulin–transferrin–selenium-X, 1 mM sodium pyruvate, 0.05 mM β-mercaptoethanol, 5 ng/mL fibroblast growth factor (FGF)-2 (Thermofisher Scientific, Waltham, MA, USA) and 5 ng/mL platelet-derived growth factor (PDGF)-AB (Peprotech, Cranbury, NJ, USA). The conditioned medium was size-fractionated using tangential flow filtration and concentrated 50 × using a membrane with a 100-kDa molecular weight cut-off (Sartorius, Göttingen, Germany). MSC exosome preparation was assayed for protein concentration using a Coomassie Plus protein assay (Thermofisher Scientific, USA), particle size concentration and distribution by ZetaView (Particle Metrix GmbH, Munich, Germany), and CD73/ecto-5′-nucleotidase (NT5E) activity using a PiColorLock Gold Phosphate Detection System (Innova Biosciences, Cambridge, UK) in accordance with the Minimal Information for Studies of Extracellular Vesicles (MISEV) [29,30], and specifically with the identity and potency metrics proposed for MSC-EV preparations [31,32]. This protocol has been used for preparation of more than 100 batches of MSC exosomes with high reproducibility. For this study, batch AC108p has a protein concentration of 1.490 mg/mL, particle concentration of 1.47 × 10^11^ particles/mL, particle modal size of 135.2 nm (Appendix A), and CD73/NT5E activity of 20.47 mU/µg protein. MSC exosomes were 0.22-µm filtered and stored in a −20 °C freezer until use.

### 2.2. Preparation of Implants

Medical grade poly (ε-caprolactone) tricalcium phosphate (PCL–TCP) (80:20%) rectangular shaped bioresorbable scaffolds with a 0/90 degrees laydown pattern, an average pore size of 1.2 mm, and 15 × 6 × 2 mm were purchased from Osteopore International Pte Ltd. All PCL–TCP scaffolds were first treated with 5 M sodium hydroxide at 37 °C for 3 h and then rinsed five times in phosphate-buffered saline (PBS) to improve the hydrophilicity and cell adhesion. Absorbable collagen sponge (ACS; Medtronic Sofamor Danek, Memphis, TN, USA) was cut into 2 mm^3^ cubes and then packed into the PCL–TCP scaffolds under aseptic conditions. Packed PCL–TCP scaffold was loaded with 100 µg MSC exosomes in 100 µL PBS or equivalent volume of PBS and incubated for 15 min at ambient temperature before implantation. The PCL–TCP scaffold serves to provide structural support to withstand any compression that could lead to extrusion of MSC exosomes from the collagen sponge. The PCL–TCP scaffold has a honeycomb microstructure that facilitates blood vessel infiltration and new bone formation and has been used in bone regeneration and spinal fusion studies [33,34].

### 2.3. Rat Posterolateral Spinal Fusion Surgery

All animal-related procedures were approved by the Institutional Animal Care and Use Committee at the National University of Singapore under protocol number R21-0567. Twenty-three 12-week-old male Sprague-Dawley rats with a mean weight of 393 ± 53 g, ranging from 340 to 446 g, were used. Since the female estrous cycle could introduce unexpected variables, only male rats were used in this study.

L4–L5 posterolateral spinal fusion surgery was performed as previously described [33,34]. Briefly, rats were anesthetized with isoflurane (5% for induction, 2% for maintenance). A posterior midline skin incision was made. Two paramedian fascia incisions were made followed by subperiosteal dissection of muscles from the transverse processes of L4 and L5 vertebrae. The transverse processes were decorticated meticulously with a scalpel. Rats were then randomly allocated into two groups: (1) Exosome group: PCL–TCP scaffold packed with ACS containing 100 µg exosomes (*n* = 12); and (2) Control group: PCL–TCP scaffold packed with ACS containing 100 µL PBS (*n* = 11). The dose of exosomes was selected based on our previous study in which a single dose of 100 µg exosomes delivered in collagen sponge was sufficient to sustain repair and regeneration of calvaria bone defects [24]. The number of animals used was based on a power analysis using our best approximation of the true effect and anticipated sample variability using the following assumption: Type I error = 0.05, and power = 0.80. Based on these calculations, we concluded that a minimum number of 11 animals was required in each group.

PBS or exosome-supplemented PCL–TCP scaffold was placed above the decorticated transverse processes and secured by 4-0 Tricon sutures (Medtronic). The fascicles and skin incision were closed with 3-0 biodegradable sutures and 3-0 non-absorbable sutures, respectively. The animals were fed *ad libitum* and allowed to move inside the cage without restriction. Post-operative buprenorphine and enrofloxacin were given for 5 days, and the animals were monitored daily for the first week. At 2 and 6 weeks post-operation, aseptically filtered solutions of calcein green (10 mg/kg) and alizarin red (25 mg/kg), utilizing a 0.22 µm syringe filter, were injected subcutaneously into the animals for measurement of the bone formation and mineral apposition rate. At 8 weeks post-operation, the animals were euthanized and the spine specimens were harvested for various analyses.

### 2.4. Manual Palpation

Fusion of the harvested spines was evaluated by two blinded observers (W.M.R.L. and L.Y.). If there was no motion present in all six directions (flexion, extension, left and right lateral bending, and left and right rotation), the segment would be graded as fused. Otherwise, it would be graded as non-union. Fusion rate was calculated by dividing the number of spines that displayed no motion in all six directions by the total number of spines and expressed as a percentage.

### 2.5. Micro-Computed Tomography Analysis

Live micro-computed tomography (µ-CT) was performed at 4 weeks post-operation at a resolution of 59 µm voxel (Quantum FX, PerkinElmer, Waltham, MA, USA). Similarly, spine specimens harvested at 8 weeks were scanned at a resolution of 59 µm voxel. The total volume of interest was defined as the space inside the scaffold and between the transverse processes. An optimized density threshold was applied to isolate mineralized bone for measurement of the bone histomorphometric parameters. Using CTAn version 1.15.4.0 (SkyScan, Kontich, Belgium), three-dimensional (3D) microstructural parameters of bone tissue were calculated, including bone volume over total volume (BV/TV, %), bone surface over bone volume (BS/BV, 1/mm), bone surface over total volume (BS/TV, 1/mm), trabecular thickness (Tb.Th, µm), trabecular number (Tb.N, 1/mm) and trabecular separation (Tb.Sp, µm). The volume of ectopic bone outside the scaffold (mm^3^) was calculated by measuring the new bone formed on facet and spinous process using the polygon tools in CTAn. By applying an optimized density threshold, the bone volume inside the selected region was calculated. Based on the µ-CT images, the fusion score was calculated using a 3-point scale: 0, no fusion; 1, unilateral fusion along the transverse process; 2, bilateral fusion along the transverse processes.

### 2.6. Histological Analysis

Tissue samples were fixed in 4% paraformaldehyde for 2 weeks and decalcified in 30% formic acid for 2 weeks. After decalcification, the samples were dehydrated, paraffin-embedded, and sectioned at 5 µm thickness using a microtome. Following standard protocols, the sections were stained with hematoxylin and eosin (HE), Masson-Goldner trichrome (MT) and Alcian blue (AB), and imaged using an inverted microscope (Nikon Eclipse Ti2, Nikon, Tokyo, Japan). Following a modified version of the histological grading scale of Emery et al. [35], a scale of 0–3 points was used by two scorers (W.M.R.L. and L.Y.) to evaluate the new bone formation using MT-stained slides. Samples were scored as follows: 0- empty cleft; 1- only fibrous tissue; 2- more fibrous tissue than bone; and 3- more bone than fibrous tissue. The percentage of fibrous tissue stained green by MT between the transverse processes was also analyzed using the ImageJ software (version 1.54i) (National Institutes of Health, Bethesda, MD, USA). Immunohistochemical staining for CD31 (ab182981, Abcam, Cambridge, MA, USA) and osteocalcin (OCN, 23418-1-AP, Thermofisher Scientific) was performed using the VECTASTAIN^®^ Elite ABC-HRP Kit (Vector Laboratories, Burlingame, CA, USA) to assess the blood vessels and bone matrix, respectively. Five fields between the transverse processes were examined under 20× magnification. CD31^+^ stained vessels were counted and expressed as CD31^+^ vessels per area (mm^2^). OCN^+^ stained area was normalized to the total area and expressed as % OCN^+^ stained area.

Resin embedding was performed for samples to measure the area fraction of new bone tissue and mineral apposition rate. Briefly, samples were dehydrated by gradient ethanol and embedded in Technovit^®^ 7100 resin (Kulzer Technik, Wehrheim, Germany). Longitudinal sections were cut with a high-speed precision saw (EXAKT 300, EXAKT Technologies, Oklahoma City, OK, USA) and grounded into sections of 25 µm thickness. Images were taken using the inverted microscope, and ImageJ software was used to measure the area fraction of new bone tissue and mineral apposition rate. The area fraction of new bone tissue was calculated by dividing the fluorochrome-labeled area by the tissue area and expressed as percentage (%). The mineral apposition rate was calculated by dividing the width (μm) between calcein green and alizarin red by the time interval (28 days) and expressed as μm/day.

### 2.7. Biomechanical Testing

Biomechanical testing was performed as previously described with slight modification [36]. The stiffness of the spine was determined by measuring the displacement distance in a cantilever system using Instron 5543 (INSTRON, Norwood, MA, USA). After thawing the frozen rat spines to room temperature, the remaining muscular tissue was removed, leaving the surrounding ligaments intact. The cranial and caudal vertebral bodies were fixed with small screws. The exposed ends of screws were embedded firmly in dental cement, polymethylmethacrylate. The specimens were checked under X-ray to prevent the screw entering the intervertebral disc. The moment was fixed at 22 Nmm, and the moment arm was fixed at 22 mm. A force of 1 N was applied to the cement block attached to the L4 vertebral body. The samples were tested in flexion, extension, and left/right lateral bending at a constant loading rate of 0.05 mm/s. Between two loadings, a relaxation time of 30 s was adopted. Three consecutive cycles were tested, and only the last one was used for calculation of stiffness. Stiffness is expressed as Nmm/degree, and range of motion under flexion, extension, and lateral bending is expressed in degrees.

### 2.8. Statistical Analysis

Data are presented as mean ± standard deviation. The distribution of data sets was checked for normality using a Shapiro–Wilk test. For biomechanical data, bone histomorphometry, histological grading, fusion score and mineral apposition rate, a two-sample *t*-test was performed for normally distributed data, whereas a Mann–Whitney test was used for non-normally distributed data. Fusion rate was evaluated by Fisher’s exact test. All statistical analyses were performed using IBM SPSS Statistics software (version 23). *p* < 0.05 was considered statistically significant.

## 3. Results

### 3.1. Manual Palpation and µ-CT Findings

Manual palpation of the harvested spines showed superior fusion rate of 83.3% in the exosome group as opposed to 27.3% in the control group at 8 weeks (*p* = 0.012) (Table 1). The fusion score was calculated based on the μ-CT images (Table 1). At 4 weeks, the exosome group had a higher radiographic fusion score of 0.58 ± 0.67 than the control group with 0.09 ± 0.30 (*p* = 0.091). At 8 weeks, the exosome group improved its fusion score to 1.42 ± 0.79, compared to the control group with 0.45 ± 0.69 (*p* = 0.013), suggesting that treatment with MSC exosomes accelerated bony fusion. To evaluate the amount and quality of new bone formation, µ-CT was performed at the L4–L5 posterolateral side of the harvested spines at 4 and 8 weeks, postoperatively (Figure 1A).

At 4 weeks, 8.3% of rats in the exosome group achieved bilateral fusion of the transverse processes, 41.7% unilateral fusion and 50% non-union, as compared to 0% bilateral fusion, 9.1% unilateral fusion and 90.9% non-union in the control group (Figure 1B). At 8 weeks, bone bridging in the exosome group continued to improve and was consistently better than in the control group. Specifically, 58.3% of rats in the exosome group achieved bilateral fusion of the transverse processes, 25% unilateral fusion and 16.7% non-union, as compared to 9.1% bilateral fusion, 27.3% unilateral fusion and 63.6% non-union in the control group (Figure 1B).

Relatively low amounts of ectopic bone were observed outside the scaffold in both exosome and control groups (7.42 ± 4.52 mm^3^ vs. 7.81 ± 3.41 mm^3^, *p* = 0.82), with no statistically significant difference (Figure 1C).

To quantify the μ-CT results, bone structural parameters including BV/TV, BS/BV, BS/TV, Tb.Th, Tb.Sp and Tb.N were analyzed (Figure 2). At 4 weeks, there were no statistically significant differences between the exosome and control groups in any of the bone structural parameters. Differences in bone parameters were more evident at 8 weeks. At 8 weeks, the exosome group had significantly higher BV/TV (10.66 ± 4.56% vs. 6.66 ± 3.42%, *p* = 0.028), BS/TV (1.09 ± 0.41/mm vs. 0.73 ± 0.26/mm, *p* = 0.018) and Tb.N (0.24 ± 0.09/mm vs. 0.14 ± 0.05/mm, *p* = 0.003). Relative to the control group, the exosome group also had lower Tb.Sp (1310 ± 261 µm vs. 1454 ± 210 µm, *p* = 0.161), albeit not statistically significant.

Additionally, the exosome group displayed a general improving trend in the bone structural parameters, with statistically significant improvements in BV/TV (4w: 5.96 ± 3.96% vs. 8w: 10.66 ± 4.56%, *p* = 0.010), BS/BV (4w: 19.20 ± 3.72/mm vs. 8w: 11.16 ± 1.60/mm, *p* < 0.001) and Tb.Th (4w: 292 ± 50 µm vs. 8w: 440 ± 84 µm, *p* < 0.001) from 4 to 8 weeks.

### 3.2. Histological Findings

The µ-CT findings were further supported by the histological results at 8 weeks (Figure 3). The PCL–TCP scaffold remnants were dissolved during tissue processing, as indicated by the void spaces in the histological images. Relative to the control group, the exosome group had more marrow cavities and more newly formed bone islands bridging the transverse processes (Figure 3A). Based on the MT-stained images, semi-quantitative analysis of the green-stained areas further confirmed that there were reduced fibrous tissues between the transverse processes in the exosome group compared to the control group (16.59 ± 9.63% vs. 29.78 ± 15.48%, *p* = 0.022) (Figure 3D).

Eight of 12 samples in the exosome group showed continuous bridging between the transverse processes, indicating complete fusion. In contrast, only 2 out of 11 samples achieved fusion in control group (Appendix A). The histological fusion rate was 66.7% in the exosome group and 18.2% in the control group (*p* = 0.024) (Figure 3B).

Based on the modified histological grading scale of Emery et al. [35], 66.6% of samples in the exosome group was scored as 3 (more bone than fibrous tissue), and the remaining 33.3% of samples scored as 2 (more fibrous tissue than bone) (Figure 3C). In contrast, 9.1% of samples in the control group was scored as 3, 45.5% scored as 2, and 45.5% scored as 1 (fibrous tissue only). Consequently, the histological score of the exosome group was significantly higher than its control counterpart (2.7 ± 0.5 vs. 1.6 ± 0.7, *p* = 0.002) (Figure 3E). 

Further analyses by immunohistochemistry revealed a significantly higher percentage of areal deposition of osteocalcin (OCN) in the exosome group than in the control group (1.19 ± 0.66% vs. 0.59 ± 0.37%, *p* = 0.014) (Figure 4A). CD31 staining to assess neovascularization also showed significantly higher density of blood vessels in the exosome group than in the control group (50.25 ± 9.80/mm^2^ vs. 37.06 ± 8.14/mm^2^, *p* = 0.002) (Figure 4B). 

To evaluate the bone formation and bone growth rate, area fractions of new bone tissues and mineral apposition rates were compared between the exosome and control group (Figure 5A–C). There was a greater area fraction of new bone tissue at 2 weeks (1.21 ± 1.23% vs. 0.86 ± 1.36%, *p* = 0.619) and 6 weeks (3.71 ± 2.02% vs. 2.85 ± 0.70%, *p* = 0.305) (Figure 5B), and therefore higher mineral apposition rate (2.28 ± 0.81 µm/day vs. 1.67 ± 0.36 µm/day, *p* = 0.097) (Figure 5C) in the exosome group than in the control group, although these differences were not statistically significant.

### 3.3. Biomechanical Findings

Biomechanical testing of the harvested spines at 8 weeks further showed improvements in stiffness with exosome treatment (Figure 6). Relative to the control group, the exosome group had ~74% increase in stiffness under flexion (41.17 ± 17.01 Nmm/Deg vs. 23.66 ± 9.57 Nmm/Deg, *p* = 0.11), ~91% increase under extension (32.91 ± 8.59 Nmm/Deg vs. 17.19 ± 8.11 Nmm/Deg, *p* = 0.033) and ~25% increase under lateral bending (44.99 ± 18.82 Nmm/Deg vs. 35.97 ± 16.83 Nmm/Deg, *p* = 0.24).

Consequently, the range of motion was significantly reduced under extension with 0.6110 ± 0.1994° in the exosome group, as compared to 1.3709 ± 0.6005° in the control group (*p* = 0.031) (Table 2). These biomechanical findings corroborated with the µ-CT and histological findings that MSC exosome treatment enhanced bone bridging between the transverse processes to facilitate spinal fusion and improve stability.

## 4. Discussion

The primary finding of the present study was that a single implantation of MSC exosome-supplemented PCL–TCP scaffold improved spinal fusion and mechanical stability through enhanced bone bridging between the transverse processes, with minimal ectopic bone formation. Of note, we did not observe any adverse responses in the immunocompetent animals used in the present study, suggesting potential allogeneic applications of human MSC exosomes.

We previously reported that MSC exosomes were efficacious in repairing osteochondral lesions and bone defects [22,24]. However, it was not clear if such therapeutic efficacy extends to spinal fusion. Using a well-established rat posterolateral spinal fusion model [33], we have demonstrated in this proof-of-concept study that a single implantation of exosome-supplemented collagen sponge packed in PCL–TCP scaffold enhanced spinal fusion and improved mechanical stability by inducing bone formation and bridging between the transverse processes, as evidenced by significant improvements in fusion score and rate, bone structural parameters, histology, stiffness, and range of motion. Along with enhanced fusion, increased vascularization, and bone matrix deposition were also observed. Notably, the fusion scores in the exosome group were 0.58 and 1.42, as opposed to the control group with 0.09 and 0.45 at 4 and 8 weeks, respectively. These findings suggest a robust and faster fusion with exosome treatment. 

In this study, we administered MSC exosomes in absorbable collagen sponge (ACS) packed in a PCL–TCP scaffold, similar to the administration of rhBMP-2 in ACS (Infuse™, Medtronic) in spinal fusion surgeries. As assessed by manual palpation at 8 weeks, the fusion rate was 83.3% in the exosome group compared to 27.3% in the control group. However, this fusion rate of 83.3% by MSC exosomes (100 µg) was lower than that of rhBMP-2 (10 µg) applied in ACS at 100%, as previously reported using the same rat posterolateral spinal fusion model [33]. Despite a relatively lower fusion rate, the volume of ectopic bone by MSC exosomes at ~7.4 mm^3^ was ~69% lower than that of rhBMP-2 at 24 mm^3^ [33]. It was further noted that compared to the amount of bone formed within the scaffold and between the transverse processes, the amount of ectopic bone with exosome treatment was relatively lower. Since there were no statistically significant differences in the amount of ectopic bone between the exosome and control groups, the ectopic bone formation could be inherent to the surgical procedure unrelated to the use of exosomes. Demineralized bone matrix (DBM) is also commonly used as bone graft in spinal fusion surgeries [37]. However, DBM has limited osteoinductive properties. A recent study evaluated several commercially available DBM products in an athymic rat posterolateral fusion model and observed large variation in the fusion at 4 weeks [38]. On this note, exosome-supplemented PCL–TCP scaffold could be a viable strategy to overcome some of the issues/challenges with use of rhBMP-2 and/or DBM.

The present study did not examine the mechanisms underlying the effects of MSC exosomes in enhancing bone formation in spinal fusion. In our previous study, we reported that MSC exosomes work through a multifaceted mechanism of enhancing M2 over M1 macrophage polarization, cell survival, proliferation, angiogenesis and osteogenesis, while suppressing apoptosis and inflammation to promote bone regeneration [24]. Some of these cellular activities mediated by MSC exosomes could be attributed to exosomal CD73-mediated adenosine receptor activation of AKT and ERK signaling [22,39]. As there were also increased neovascularization and bone matrix deposition, along with enhanced bone formation observed with exosome treatment, it is likely that MSC exosomes work through a similar multifaceted mechanism to promote bone formation in spinal fusion. Regarding the process of spinal fusion, intramembranous ossification represents the primary pathway in bone formation [40]. However, endochondral ossification also plays a prominent role in spinal fusion. In this study, we observed cartilage islands between the transverse processes (Appendix A), in agreement with studies that reported cartilage as part of the early fusion mass [40,41]. As MSC exosomes demonstrate efficacy for cartilage and bone repair [22,24], it is likely that these nanovesicles augment cartilage and bone biology at multiple levels of gene regulation, cell activation and tissue morphogenesis to positively affect the fusion success.

This study has some limitations. Histological analysis was performed at only one timepoint (8 weeks). Future studies should include multiple timepoints to examine the effects of MSC exosomes over time, the kinetics of fusion and the bone-formation process (i.e., intramembranous vs. endochondral ossification). Only a single dose of MSC exosomes and one scaffold type were tested. Further optimization of the exosome dose and scaffold delivery will be required to improve spinal fusion. Future studies are needed to determine the biodistribution of exosomes and their persistence at the fusion site. Studies using female rats may be considered to determine any gender differences. Although the fusion efficacy with the use of MSC exosomes is promising, further validation in a clinically relevant large animal model is needed for clinical translation.

## 5. Conclusions

In this study, we have shown that MSC exosomes delivered in collagen sponge packed in PCL–TCP scaffold have the capacity to improve spinal fusion and stability through enhanced bone bridging between the transverse processes, with minimal ectopic bone formation and no adverse responses. Our findings provide the basis for development of MSC exosomes as a cell-free therapeutic for supplementation to clinically used bone grafts for enhanced spinal fusion.

## Figures and Tables

**Figure 1 cells-13-00761-f001:**
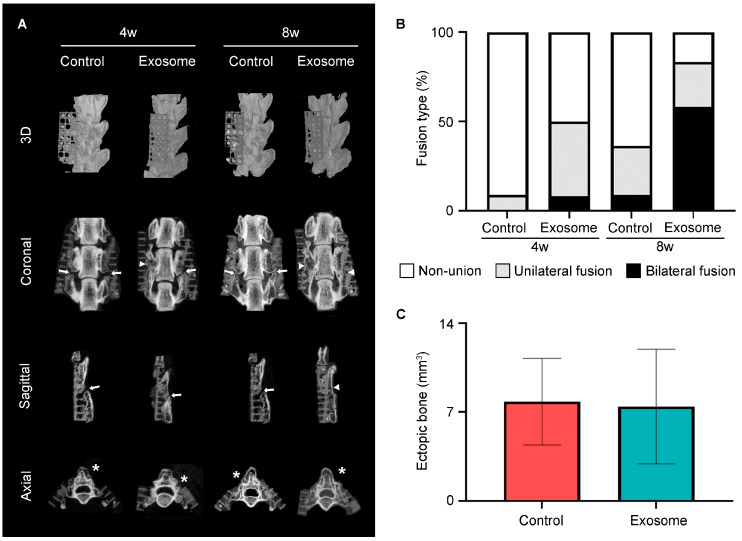
Micro-CT evaluation of spinal fusion at 4 and 8 weeks post-surgery. (**A**) Representative μ-CT images of reconstructed 3D, coronal, sagittal and axial views of the L3–L5 spinal segment. Clear gaps between the L4 and L5 transverse processes are indicated by white arrows. Successful bony bridges are indicated by white triangles. Ectopic bone is indicated by asterisks. (**B**) Percentage of rats showing bilateral fusion, unilateral fusion, or non-union. (**C**) Volume of ectopic bone (mm^3^) in control and exosome-treated rats at 8 weeks post-surgery. Control group (*n* = 11). Exosome group (*n* = 12). Results are presented as mean ± standard deviation.

**Figure 2 cells-13-00761-f002:**
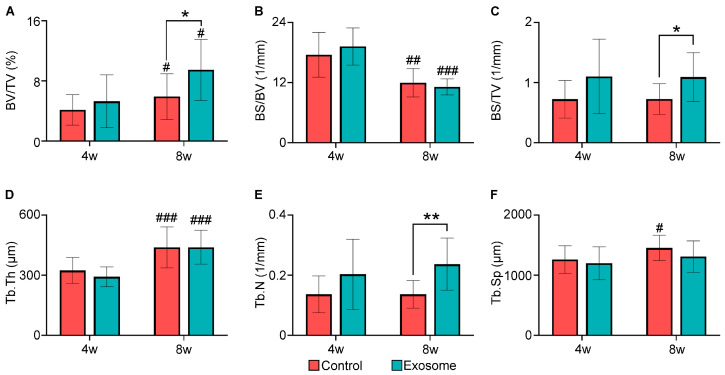
Bone histomorphometric parameters of bone formation at 4 and 8 weeks post-surgery. Microarchitecture analysis of bone mass inside the scaffolds and between the transverse processes was performed. (**A**) Bone volume over total volume (BV/TV, %), (**B**) bone surface over bone volume (BS/BV, 1/mm), (**C**) bone surface over total volume (BS/TV, 1/mm), (**D**) trabecular thickness (Tb.Th, µm), (**E**) trabecular number (Tb.N, 1/mm) and (**F**) trabecular separation (Tb.Sp, µm). Control group (*n* = 11). Exosome group (*n* = 12). Results are presented as mean ± standard deviation. * *p* < 0.05 and ** *p* < 0.01 compared to control group. # *p* < 0.05, ## *p* < 0.01, and ### *p* < 0.001 compared to 4 weeks.

**Figure 3 cells-13-00761-f003:**
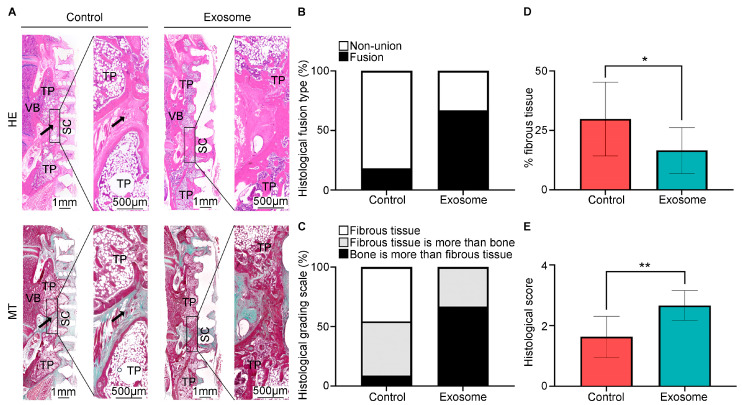
Histological evaluation of spinal fusion at 8 weeks post-surgery. (**A**) Representative photomicrographs of hematoxylin and eosin (HE) and Masson-Goldner trichrome (MT) stained sections. (**B**) Histological fusion type, (**C**) histological grading scale, (**D**) percentage of fibrous tissue and (**E**) histological scores comparing the control group (*n* = 11) and exosome group (*n* = 12). Histological non-union, presented as fibrous tissue between the L4 and L5 transverse process (TP), are indicated by black arrows. Scale bars indicate 1 mm or 500 µm. Results are presented as mean ± standard deviation. * *p* < 0.05 and ** *p* < 0.01 compared to control. VB: vertebral body, SC: scaffold.

**Figure 4 cells-13-00761-f004:**
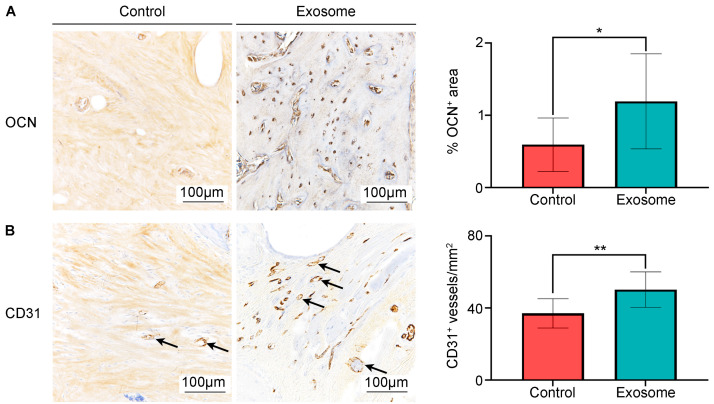
Evaluation of neovascularization and bone matrix deposition at 8 weeks post-surgery. (**A**) Osteocalcin (OCN) staining and semi-quantitative analysis of OCN^+^-stained areas. (**B**) CD31 staining and semi-quantitative analysis of CD31^+^ microvessels. The microvessels are indicated by the black arrows. Representative images. Scale bars indicate 100 µm. Control group (*n* = 11). Exosome group (*n* = 12). Results are presented as mean ± standard deviation. * *p* < 0.05 and ** *p* < 0.01 compared to control.

**Figure 5 cells-13-00761-f005:**
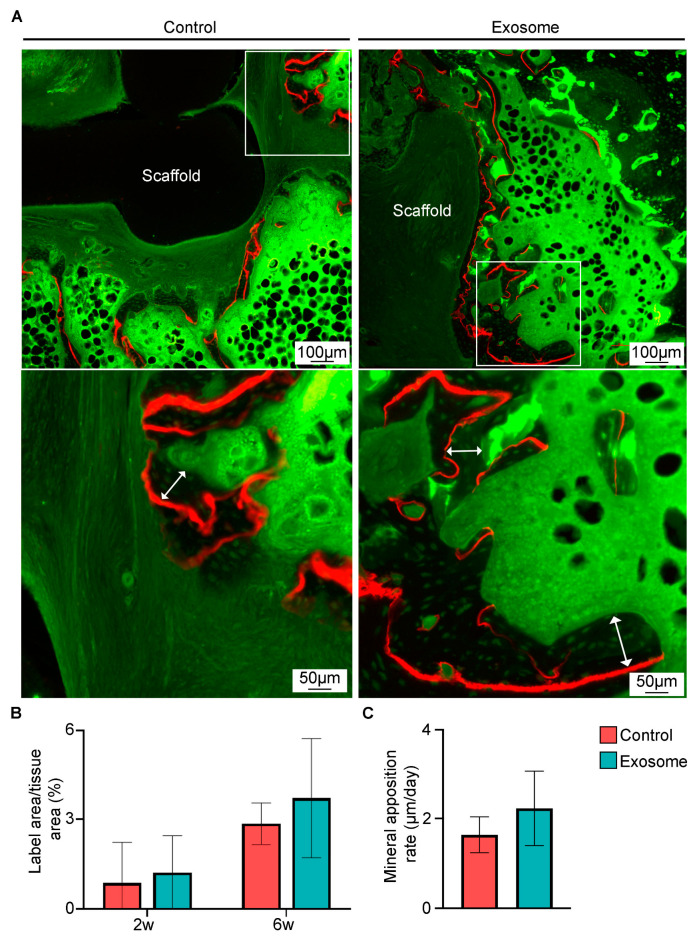
Evaluation of new bone growth during spinal fusion. (**A**) Representative calcein green and alizarin red labeled images taken on resin sections (*n* = 7 per group). Scale bars indicate 100 µm or 50 µm. Using the fluorochrome labeled images, measurements of the (**B**) area fraction of new bone tissue at 2 and 6 weeks and (**C**) mineral apposition rate were performed. The mineral apposition rate was calculated by dividing the distance as indicated by the double-headed arrow between green (calcein green) and red (alizarin red) by the time interval (28 days). Data are presented as mean ± standard deviation.

**Figure 6 cells-13-00761-f006:**
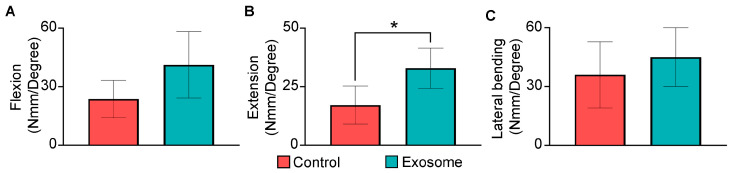
Biomechanical stiffness analysis to assess segmental stability of the operated lumbar spine at 8 weeks. Stiffness in (**A**) flexion, (**B**) extension and (**C**) lateral bending were compared between the control group (*n* = 4) and exosome group (*n* = 5). Data are presented as mean ± standard deviation. * *p* < 0.05 compared to control.

**Table 1 cells-13-00761-t001:** Radiographic fusion score and fusion rate.

	Control (*n* = 11)	Exosome (*n* = 12)	*p* Value
Fusion score (4w)	0.09 ± 0.30	0.58 ± 0.67	0.091
Fusion score (8w)	0.45 ± 0.69	1.42 ± 0.79	0.013
Fusion rate (8w)	27.3%	83.3%	0.012

Based on μ-CT images, fusion score was calculated using a 3-point scale: 0, non-union; 1, unilateral fusion; and 2, bilateral fusion along the transverse processes. Fusion rate was determined by manual palpation.

**Table 2 cells-13-00761-t002:** Range of motion.

	Control (*n* = 4)	Exosome (*n* = 5)	*p* Value
Flexion	0.9352 ± 0.3038°	0.6006 ± 0.2424°	0.0539
Extension	1.3709 ± 0.6005°	0.6110 ± 0.1994°	0.031
Lateral bending	0.6755 ± 0.2979°	0.4940 ± 0.1753°	0.0781

## Data Availability

Data are available upon reasonable request to the corresponding authors.

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
