# Peer review of "Mesenchymal Stem Cell Exosomes Enhance Posterolateral Spinal Fusion in a Rat Model"

_cells, 2024, doi:10.3390/cells13090761_

Round 1
Reviewer 1 Report
Comments and Suggestions for Authors
The manuscript, “Mesenchymal stem cell exosomes enhance posterolateral spinal fusion in a rat model” is a well written study that clearly demonstrates the usefulness of MSC derived exosomes for spinal fusion. Using a rat model of spinal fusion, the MSC derived exosomes enhanced bone bridging, bone mechanical properties, and bone formation rate. It is a nice and straight forward study with only a few comments:
1. It is not clear why only male rats were used in this study.
2. Was a power analysis performed to determine sample size?
3. It is appreciated that the discussion includes another study that used BMP2. What do the authors think of the exosomes compared to demineralized bone matrix? Are there any studies that could also be included into the discussion to help compare outcomes of the various materials? Are there any other positive controls that could be used with this study?
4. It is not clear why 100 ug of exosomes where used. Is this based on a previous study?
5. Is intramembranous or endochondral ossification primarily observed in this model and do the exosomes affect it? It would be interesting if exosomes only alter timing or if there is a difference in the ossification process. Did previous studies investigate this (long bone defect)?
6. The authors included possible mechanisms in the discussion based on previous studies. Since histology is available, some validation should be included in the study. Also is the difference in bone bridging due to difference in osteoblast numbers or activity of the osteoblasts?
Reviewer 2 Report
Comments and Suggestions for Authors
This manuscript explored the therapeutic effects of MSC exosomes in spinal fusion using an absorbable collagen sponge packed in a poly Ɛ-caprolactone tricalcium phosphate (PCL-TCP) scaffold as the delivery vehicle in a rat posterolateral spinal fusion model. While the authors reported positive effects of the MSC exosomes, several issues need to be addressed to enhance the quality of the study:
1) Line 111, Page 3: the authors should clarify the body weight of rats used in this study, 384±29 g would be 355-413 g but not 340-446g.
2) Although publications related to E1-MYC 16.3 human MSC line were cited in this manuscript, the basic characterization of the cells and more importantly, of the exosomes (such as morphology, diameter, surface marker, etc.) should be provided in this article.
3) Although HE staining and Masson-Goldner trichrome (MT) staining images can reveal new bone formation, other immunohistochemical/immunofluorescence staining should be included in this study.
4) The animal models of rat posterolateral spinal fusion surgery usually lead to higher lethality than other animal models, if possible, the authors could measure the survival curves.
5) Geometry and architecture of implants like PCL-TCP used in this manuscript can profoundly influence the therapeutic effect. Thus, the authors should describe the shape and architectural features of the materials they used in this study. Providing SEM images would be much better.
Comments on the Quality of English Language
Quality of English Language is fine
Reviewer 3 Report
Comments and Suggestions for Authors
Please see attached document

Minor editing of English language required
Round 2
Reviewer 2 Report
Comments and Suggestions for Authors
The authors have addressed all my comments.